# Universally bistable shells with nonzero Gaussian curvature for two-way transition waves

Nikolaos Vasios[1], Bolei Deng[1], Benjamin Gorissen [1] & Katia Bertoldi [1✉]

Multi-welled energy landscapes arising in shells with nonzero Gaussian curvature typically fade away as their thickness becomes larger because of the increased bending energy required for inversion. Motivated by this limitation, we propose a strategy to realize doubly curved shells that are bistable for any thickness. We then study the nonlinear dynamic response of one-dimensional (1D) arrays of our universally bistable shells when coupled by compressible fluid cavities. We find that the system supports the propagation of bidirectional transition waves whose characteristics can be tuned by varying both geometric parameters as well as the amount of energy supplied to initiate the waves. However, since our bistable shells have equal energy minima, the distance traveled by such waves is limited by dissipation. To overcome this limitation, we identify a strategy to realize thick bistable shells with tunable energy landscape and show that their strategic placement within the 1D array can extend the propagation distance of the supported bidirectional transition waves.

[1] John A. Paulson School of Engineering and Applied Sciences, Harvard University, Cambridge, USA. ✉email: bertoldi@seas.harvard.edu

Curved elastic shells have drawn significant interest, not only because of their outstanding structural performance but also for their extraordinarily rich nonlinear behavior[1–10]. In particular, curved elastic shells with low thickness to radius ratio typically possess two stable configurations[1,6,11,12]—a feature that has been exploited to realize tunable lenses[13], as well as valves for autonomous control of soft actuators[14]. However, the low thickness to radius ratio of such shells makes them extremely sensitive to imperfections and, therefore, limits their possible range of applications. On the other hand, curved elastic shells with large thickness to radius ratio are structurally more robust, but typically lack bistability.

Multistable structures comprising arrays of interconnected bistable elements have recently emerged as a powerful platform to manipulate and control the propagation of mechanical signals, owing to their ability to support the propagation of transition waves—nonlinear waves similar to those of falling dominoes that sequentially switch all elements[15]. Such transition waves have been recently exploited to enable unidirectional propagation[16–18], achieve complex shape reconfigurations[19] and realize structures that can be quickly deployed[20], as well as mechanical logic gates[21]. However, almost all previous studies have focused on bistable elements that possess two energy minima of different height[16–21] and, therefore, support unidirectional wave propagation. By contrast, the advantages and challenges associated with the propagation of transition waves in systems whose constituents possess equal energy minima have received very limited attention[22].

In this work, we first identify a strategy to realize bistable doubly curved shells with arbitrary thickness. We then focus on arrays of such bistable shells and use a combination of experiments and numerical simulations to study their non-linear dynamic response. Owing to their doubly–curved nature, the shells can be connected using rigid tubes, to form airtight cavities between neighboring elements. Importantly, such fluidic cavities introduce a coupling between the shells and enable the propagation of transition waves, which sequentially switch the shells from one stable configuration to the other. Here, we systematically study the propagation of transition waves in arrays of bistable shells with equal energy minima. We show that the velocity of the propagating transition waves in such systems is not a fixed system property, but can rather be tuned by controlling the energy supplied to initiate the pulses. Further, we find that the propagation of the transition waves is limited by dissipation. To overcome this limitation, we introduce curved elastic shells with tunable energy profile and demonstrate that, when few of such elements are embedded into our arrays, the waves can propagate for longer distances while maintaining bidirectionality.

## Results

**Design of thick bistable shells.** We begin by considering doubly curved thick shells (i.e., shells with non-zero Gaussian curvature) obtained by the 360° revolution of the height profile (see black dashed line in Fig. 1a)

$$h = \begin{cases} H\left[1 + 2\left(\frac{r}{R}\right)^3 - 3\left(\frac{r}{R}\right)^2\right], & r \in [0, R] \\ 0, & r \in [R, R+S], \end{cases} \quad (1)$$

where $H$ is the maximum shell height, $R$ is the shell radius and $S$ denotes the length of the flat portion added at the base of the shell to facilitate the enforcement of boundary conditions in

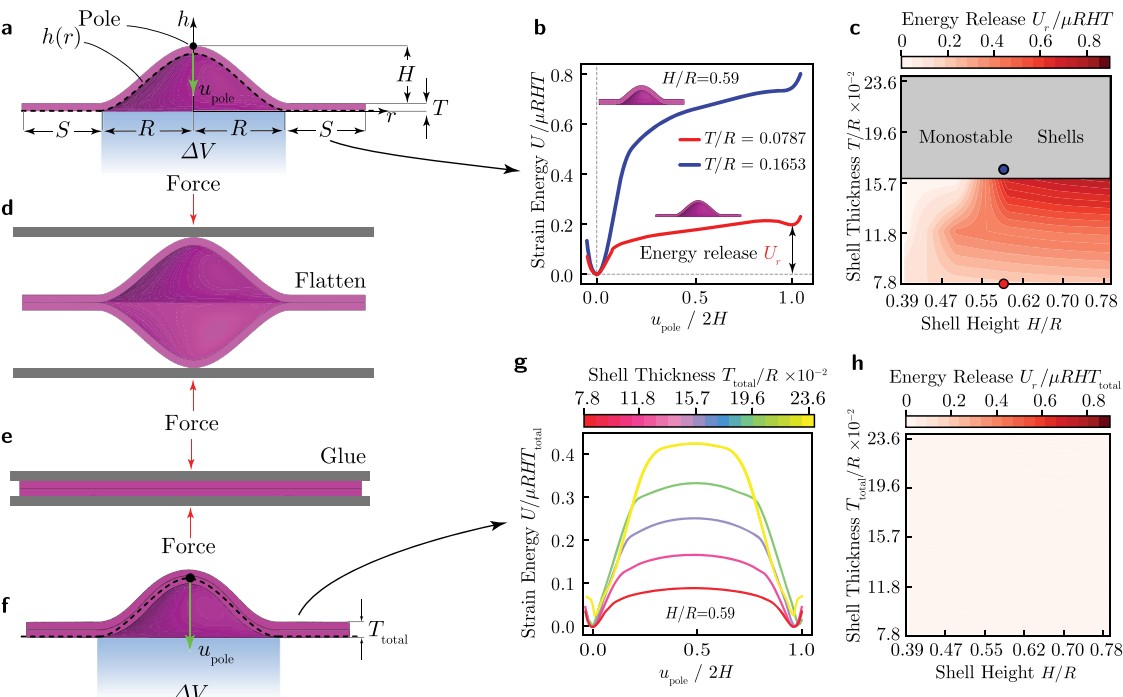

**Fig. 1 Our shells. a–c** Single shells. **a** Shell geometry, obtained by the 360° revolution of the height profile $h(r)$ (dashed line) defined in Eq. (1). Note that $H$ denotes the maximum shell height, $R$ is the shell radius, $T$ is the shell thickness and $S$ is the length of the flat portion added at the base. The blue shaded region indicates the portion of the shell that is inflated and deflated and $u_{pole}$ represents the pole displacement. **b** Elastic strain energy landscape as a function of the pole displacement during the quasi-static inflation/deflation of two shells with $H/R = 0.59$ and $T/R = 0.0787$ (red) and $T/R = 0.1653$ (blue). **c** Evolution of the energy released, $U_r$, upon inversion as a function of $H/R$ and $T/R$. The red marker corresponds to the energy release for a shell with $T/R = 0.0787$ whereas the blue marker for a shell with $T/R = 0.1653$. All shell geometries that lie in the gray shaded area are found to possess only a single stable state (the undeformed–"as fabricated" state) and are therefore monostable. **d–h** Double shells. **d** Flattening of two identical single shells. **e** Gluing the two single shells in the flat deformed configuration to obtain the double shell. **f** The geometry of the double shell, where $T_{total}$ corresponds to the total thickness of the double shell. **g** Strain energy landscape for double shells with different thickness. **h** Contour plot of the energy released, $U_r$, as a function of $H$ and $T_{total}$.

experiments and simulations. The final shell geometry, shown in Fig. 1a, is obtained by offsetting the height profile $h$ by a distance equal to the shell thickness $T$. To investigate the quasi-static response of such shells upon pressurization, we conduct Finite Element (FE) analyses using the commercial package ABAQUS 2019/Standard. In the analyses we create half shell models, mesh them using 8-node fully integrated hybrid linear brick elements (Abaqus Element code C3D8H) and use an incompressible hyperelastic Neo-Hookean material with initial shear modulus, $\mu$, to capture the material's response (see Supplementary Note 1.4 and Supplementary Figs. 9–12). Further, we impose symmetry boundary conditions and subject the models to inflation and deflation by controlling the enclosed volume through the fluid filled cavity interaction (see Fig. 1a). In Fig. 1b we report the evolution of the elastic strain energy, $U$, as a function of the pole displacement, $u_{pole}$, for two shells characterized by $H/R = 0.59$, but with $T/R = 0.0787$ (red line) and $T/R = 0.1653$ (blue line). We find that the thinner shell features an elastic strain energy landscape with two energy minima at $u_{pole} = 0$ and $u_{pole} \approx 2H$ and, therefore, is bistable. Importantly, due to finite thickness effects, the stable configuration at $u_{pole} \approx 2H$ is characterized by an energy state higher than that of the undeformed one. As a result, the shell releases energy $U_r$ when transitioning from its inverted state to its initial one (see Fig. 1b). By contrast, the strain energy landscape of the thicker shell monotonically increases with the pole displacement $u_{pole}$, indicating that the particular shell is monostable. A more systematic analysis on the effect of shell height $H$ and thickness $T$ to the response of the shells reveals that those with $T/R < 0.159$ have two stable states, whereas those with $T/R > 0.159$ are monostable (see Fig. 1c).

Next, since the results of Fig. 1c indicate that our thick doubly curved shells with $T/R > 0.159$ are monostable for any choice of $H/R$, we identify a strategy to realize shells that possess two stable states for any set of geometric parameters. To obtain such shells, we combine two identical doubly curved shells with thickness $T$, height $H$ and the profile given by Eq. (1). We first compress the two identical shells until they elastically deform into a flat configuration (see Fig. 1d) and then glue them together (see Fig. 1e). To assess the bistability of the resulting shells (see Fig. 1f), we use FE simulations in which we account for the entire gluing process (see Supplementary Note 1.4 and Supplementary Fig. 10). In Fig. 1g we report the strain energy landscape predicted by our FE analyses for shells with height $H/R = 0.59$ and total thickness $T_{total}/R \in [0.078, 0.236]$ (with $T_{total} = 2T$). Remarkably, we find that all considered shells are bistable and characterized by two strain energy minima with identical height due to the engineered stress symmetry between the inverted and initial stable states. Joining the two single shells in a deformed configuration coinciding with the horizontal symmetry plane, induces a residual stress field (symmetric about the horizontal plane) in the joint double shell, which ensures that the resulting double shell will be bistable. Finally, in Fig. 1h we report the evolution of the energy release $U_r$ as a function of both $H/R$ and $T_{total}/R$ for $T_{total}/R \in [0.078, 0.236]$ and $H/R \in [0.39, 0.78]$. Our results indicate that the energy release $U_r$ is zero for all the considered geometry combinations, suggesting that our double shells are bistable for any choice of height and thickness and always possess equal energy minima.

To quantify the validity of our numerical simulations we fabricate a double shell with a total thickness of $T_{total} = 4$ mm, out of silicone rubber (Elite Double 8, Zhermack–with an initial shear modulus $\mu = 83$ kPa[23]) and two identical shells with radius $H/R = 0.59$, $T/R = 0.079$, and $R = 25.4$ mm (see Supplementary Note 1.2 and Supplementary Figs. 4–5). We then characterize its quasi-static response by attaching its boundaries to an enclosed rigid cylinder and supplying water with a syringe pump (Pump

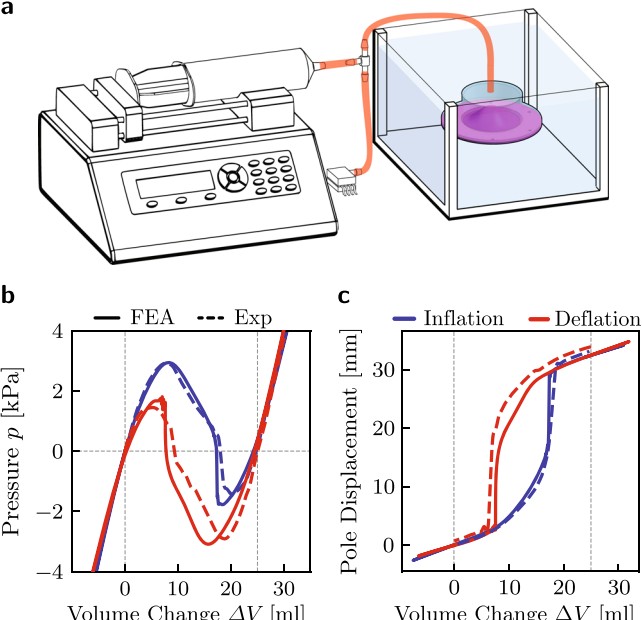

**Fig. 2 Experimental characterization of our universally bistable thick shells. a** Schematic of the experimental setup used to quasi-statically inflate and deflate the universally bistable shells using water, while being submerged in a water tank. **b**, with **c** Quasi-static pressure-volume and pole displacement-volume relationships obtained upon inflation (blue lines) and deflation (red lines) of a double shell with $H/R = 0.59$ and $T_{total}/R = 0.158$ (with $R = 25.4$ mm) in experiments (dashed lines) and FE simulations (solid lines). Vertical black lines indicate the location of the two stable states for the shell.

33DS, Harvard Apparatus) at a constant rate of 30 mL/min to inflate it and deflate it (see Fig. 2a). The pressure-volume curve of our shell is obtained by monitoring the pressure during the tests with a pressure sensor (MPXV7025DP by NXP USA), whereas to monitor the displacement of the shell's pole we recorded videos which we processed to extract the displacement history of its center point (see Supplementary Note 1.3 and Supplementary Figs. 6–8). The experimental results shown in Fig. 2b,c do not only confirm bistability (see region of negative pressure in Fig. 2b), but also indicate that the engineered stress symmetry of our shells leads to pressure-volume and pole displacement-volume curves which are entirely symmetric between loading and unloading. Further, the good agreement between the experimental and numerical data, verifies the predictive ability of our FE simulations.

**Propagation of transition waves in arrays of universally bistable shells.** Having identified a strategy to realize doubly curved shells that are bistable for any combination of geometric parameters, we now arrange our universally bistable elements in 1D arrays and study their non-linear dynamic behavior. Specifically, we focus on double shells with $H/R = 0.59$, $T_{total}/R = 0.158$ and $R = 25.4$ mm, and connect them using acrylic tube segments with length $L_t$ and internal radius equal to the radius of the shells (see Fig. 3a). When the array is assembled, each tube segment encloses a finite volume of air $V_{air} = \pi R^2 L_t$. Importantly, such finite air volumes act as nonlinear nearest neighbor springs, since any deformation of the adjacent shells causes a volume change, which generates a resistant force to the shells. As such, our system comprises a 1D array of nonlinear bistable elements (i.e., bistable shells) with nearest neighbor interactions. To study its nonlinear

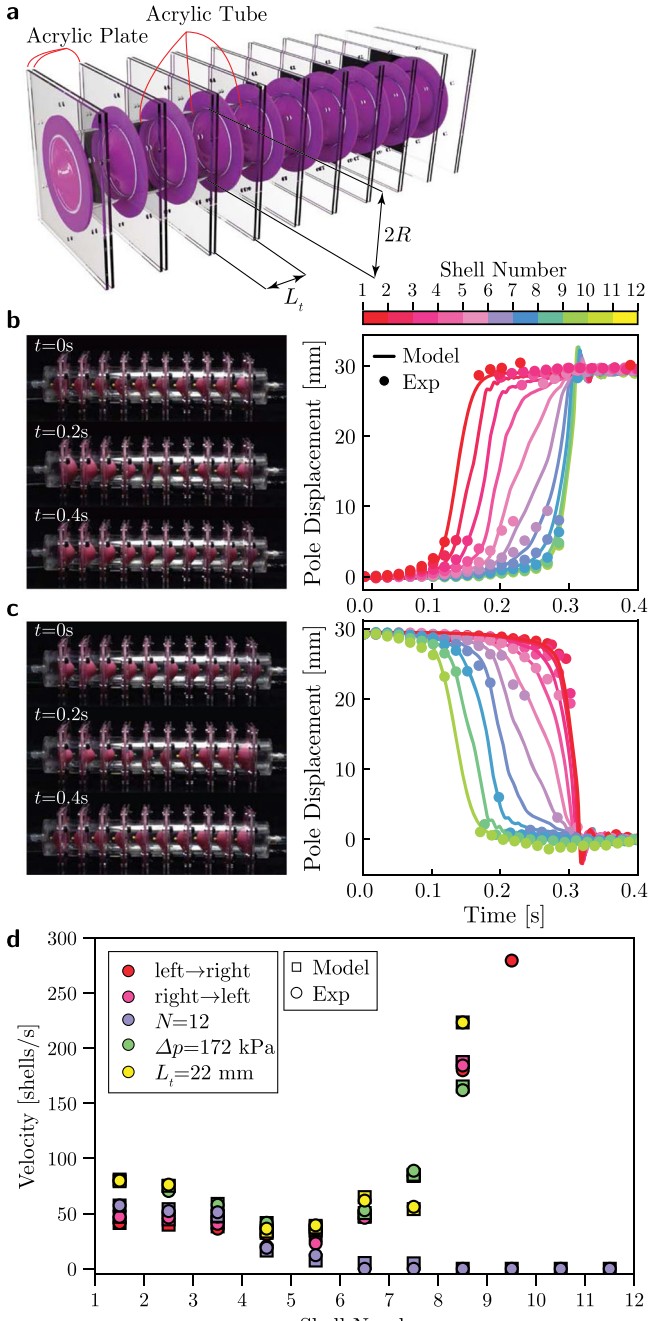

**Fig. 3 Bidirectional transition waves in 1D arrays of bistable shells connected with compressible fluid cavities. a** Schematic of the 1D array. **b,c** Bidirectional propagation of transition waves in an array of 10 universally bistable shells with $H = 15$ mm, $R = 25.5$ mm and $T_{total} = 4$ mm, excited by supplying $\Delta p = 69$ kPa of pressure for 100 ms. **d** Evolution of transition wave velocity during propagation for an array of 10 universally bistable shells excited at the left (red markers) and right (pink markers) ends by applying a pressure $\Delta p = 69$ kPa for 100 ms. Blue markers represent the velocity for an identical pulse propagating in an array of $N = 12$ universally bistable shells, whereas green and yellow markers correspond to the wave velocity for a pulse excited using $\Delta p = 172$ kPa and a pulse in an array with reduced shell to shell spacing ($L_t = 22$ mm), respectively.

dynamic response, we apply a pressure pulse (i.e., a constant pressure $\Delta p$ for 100 ms) to the first/last shell of the array (while keeping the other end at atmospheric pressure) and monitor the propagation of the initiated pulse.

In Fig. 3b, we report results for an array comprising $N = 10$ double shells connected via 11 acrylic tubes with length $L_t = 28$ mm. We first arrange all shells with the pole pointing to the left (i.e., $u_{pole,i} = 0$ mm with $i = 1, 10$) and apply a pressure pulse with magnitude $\Delta p = 69$ kPa to the first unit on the left. We find that the applied pressure initiates a transition wave that sequentially switches all shells to their inverted state corresponding to $u_{pole,i} = 2H$. We then apply an identical pressure pulse to the last shell in the array and observe the propagation of another transition wave that sequentially resets all shells back to their initial configuration (Fig. 3c). To better characterize these elastic waves, in Fig. 3d we report the evolution of their velocity (calculated by monitoring the time at which $u_{pole,i} = H$) during propagation. We then find that the two pulses considered in Fig. 3b and c propagate with similar velocities through the array (see red and pink markers in Fig. 3d), indicating that our system supports bidirectional transition waves. While the resetting of bistable systems typically requires application of external forces[16,19–21], such bidirectionality provides a simple mechanism to bring the system back to its initial configuration. Importantly, Fig. 3d also reveals that the wave velocity is not constant during propagation, but rather "v-shaped" because of the combined effect of dissipation (introduced by both the fluid cavities and the elastomeric shells) and the free boundary. Damping progressively reduces the energy carried by the waves, thereby reducing the transition wave velocity. On the other hand, when the head of the pulse reaches the end of the array, the energy required to switch the last few units decreases, thereby leading to an increase of the transition wave velocity. It is important to note that propagation of the pulses through the entire array is only possible when dissipation and the size of the array (i.e., $N$) are carefully balanced. For instance, if in our structure $N$ is increased to 12, we find that the pulse stops after switching 6 shells, since it loses all its energy before being sufficiently close to the free end so as to benefit from boundary effects (see blue markers in Fig. 3d). Finally, the results of Fig. 3d highlight two promising strategies to tune the wave speed. First, the wave velocity can be controlled by varying the length of the connected tube segments, as this alters the effective stiffness of the nearest-neighbor springs provided by the air cavities. By reducing $L_t$ to 22 mm we find that the pulse maintains a higher velocity for longer distance and is less affected by dissipation (see yellow markers in Fig. 3d). Second, the wave velocity in our array can be tuned by controlling the energy supplied to initiate the pulses. In an array with $N = 10$ shells, we find that an increase of the magnitude of the applied input pressure from $\Delta p = 69$ kPa to $\Delta p = 172$ kPa results in a substantially faster pulse (see green markers in Fig. 3d). Note that this feature marks an important difference between our system and bistable structures with energy minima of different height, since for the latter the wave velocity is governed by the energy difference between their two stable states and, therefore, is a fixed system property given a certain geometry[16,19–21].

In order to obtain a better understanding for the dynamic response of our system and ensure that the behaviors observed in the tests are not introduced by experimental artifacts, we develop a numerical model. To this end, we focus on the $[i]$-th shell, which is connected to the $[i-1]$-th and $[i+1]$-th shell through tubes with radius $R$ and length $L_t$ (see Fig. 4a), and write its equation of motion as (see Supplementary 2.3)

$$m \frac{d^2 u_{pole,i}}{dt^2} + \beta \frac{du_{pole,i}}{dt} + \frac{dU(u_{pole,i})}{du_{pole,i}} + f_{i-1} - f_i = 0, \quad (2)$$

where $m$ is the mass of the shell, $\beta$ is a viscous damping parameter whereas $u_{pole,i}$ and $U(u_{pole,i})$ denote the pole

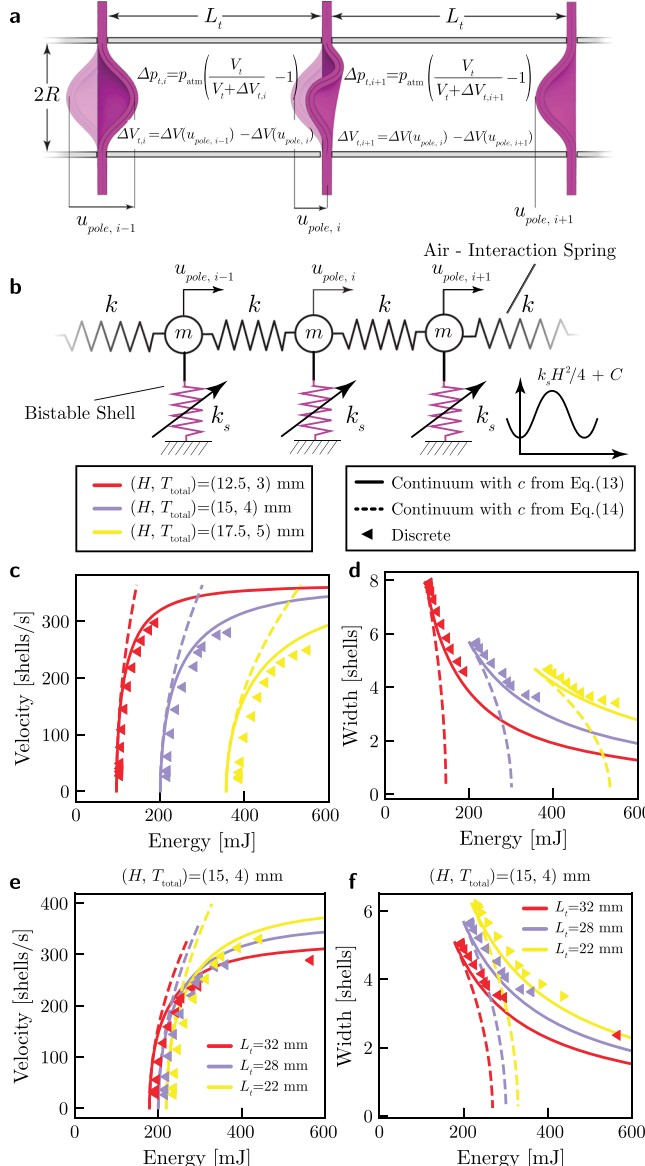

**Fig. 4 Analytical and numerical results in the absence of dissipation.**
**a** Schematic of our system, showcasing the $i - 1$, $i$ and $i + 1$ shells during the propagation of a transition waves that sequentially switches the shells from one stable state to another. **b** Discrete mass-spring model used to represent the response of our system. **c,d** Effect of the input energy provided to initiate the pulse on (**c**) the pulse velocity, $c$, and (**d**) the pulse width, $w$, for three shell geometries with $(H, T_{total}) = (12.5, 3)$ mm (yellow), $(H, T_{total}) = (15, 4)$ mm (blue) and $(H, T_{total}) = (17.5, 5)$ mm (red) and $R = 25.4$ mm, as predicted by the discrete (markers) and continuum models (lines). **e** Wave velocity, $c$, and (**f**) width, $w$, vs. input energy, $E_{in}$, for an array of universally bistable shells with $(H, T_{total}) = (15, 4)$ mm and $R = 25.4$ mm, for three values of shell-to-shell spacing, $L_t = 32$ mm (yellow), 28 mm (blue) and 22 mm (red), as predicted by the discrete (markers) and continuum model (lines). Note that in **c–f** we report two analytical solutions: one in which $c$ is obtained by solving Eq. (13) (solid lines) and one in which $c$ is given by Eq. (14) (dashed lines).

displacement and the strain energy potential of the [$i$]-th shell, respectively. Finally, $f_{i-1}$ and $f_i$ represent the interaction forces acting on the [$i$]-th shell due to the changes in volume in tubes [$i - 1$] and [$i$], respectively. Such interaction forces can be

determined using Boyle's law as,

$$f_i = \pi R^2 p_{atm} \left( \frac{\pi R^2 L_t}{\pi R^2 L_t + \Delta V_{i+1} - \Delta V_i} - 1 \right),$$
$$f_{i-1} = \pi R^2 p_{atm} \left( \frac{\pi R^2 L_t}{\pi R^2 L_t + \Delta V_i - \Delta V_{i-1}} - 1 \right),$$
(3)

where $p_{atm}$ is the atmospheric pressure, and $\Delta V_j$ is the volume change associated with the pole displacement of the [$j$]-th shell (see Supplementary 1.3). For an array comprising $N$ double shells, Eqs. (2) results in a system of $N$ coupled differential equations, which we numerically solve (using a Python implementation of the Dormand Prince 8(5,3) method[24]) to determine the pole displacement of the [$i$]-th shell as a function of time $t$.

To test the relevance of our discrete model, we first compare its predictions to the experimental results shown in Fig. 3. In all our numerical analyses we use $m = 30$ g and $\beta = 2.5$ kg/s (note that $\beta$ is determined by fitting the result from our discrete model to the experimental results of Fig. 3b and its then kept fixed for all other numerical simulations) and determine $\Delta V_i$ and $U$ associated with $u_{pole,i}$ by linearly interpolating the FE results shown in Fig. 1g–h (see Supplementary Fig. 11). Further, to ensure identical boundary conditions, we apply the experimentally extracted displacement signal to the shell from which the pulse is initiated and leave the opposite end at atmospheric pressure. We find that our numerical analyses can successfully reproduce all the experimental results reported in Fig. 3, confirming the validity of our discrete model.

Next, in an attempt to derive analytical expressions for the wave velocity, we neglect dissipative phenomena and approximate the interaction forces acting on the [$i$]-th shell as

$$f_i \approx k(u_{pole,i} - u_{pole,i+1})$$
$$f_{i-1} \approx k(u_{pole,i-1} - u_{pole,i}),$$
(4)

where $k$ is a linear approximation for the effective stiffness of the nonlinear nearest neighbor springs provided by the air cavities (see Supplementary Note 2.4 and Supplementary Fig. 23). By substituting Eq. (4) into Eq. (2) and setting $\beta = 0$, we obtain

$$m \frac{d^2 u_{pole,i}}{dt^2} + \frac{dU(u_{pole,i})}{du_{pole,i}} +$$
$$+ k(u_{pole,i+1} - 2u_{pole,i} + u_{pole,i-1}) = 0.$$
(5)

Then, we introduce a continuous function $u(\tilde{x}, t)$ that interpolates the pole displacement of [$i$]-th shell located at $\tilde{x} = x_i/L_t = i$ as $u(\tilde{x} = i, t) = u_{pole,i}$. We also assume that the width of the propagating pulses is much larger than the shell to shell distance and express $u_{pole,i\pm1}$ using Taylor expansion as

$$u_{pole,i\pm1} = u(i \pm 1, t) = \left[ u \pm \frac{\partial u}{\partial \tilde{x}} + \frac{1}{2} \frac{\partial^2 u}{\partial \tilde{x}^2} \right]_{\tilde{x}=i}.$$
(6)

Substitution of Eq. (6) into Eq. (5) yields

$$(c_0^2 - c^2) \frac{\partial^2 u}{\partial \zeta^2} = \frac{1}{m} \frac{dU(u)}{du},$$
(7)

where $\zeta = \tilde{x} - ct$ is the traveling wave coordinate, $c$ is the wave velocity and $c_0^2 = k/m$ (see Supplementary Note 2.4). Finally, to analytically solve Eq. (7) we assume that the bistable energy potential for the double shells can be approximated as

$$U(u) \approx \frac{1}{4} k_s u^2 \left( \frac{u}{H} - 2 \right)^2 + C,$$
(8)

where $C$ denotes the height of the two energy minima located at $u = 0$ and $2H$ and $0.25 k_s H^2$ is the height of the energy barrier that has to be overcome to switch the shells from one stable state to

the other. By introducing Eq. (8), Eq. (7) simplifies to

$$\frac{\partial^2 u}{\partial \zeta^2} = \frac{c_s^2}{c_0^2 - c^2} u \left( \frac{u}{H} - 1 \right) \left( \frac{u}{H} - 2 \right), \tag{9}$$

where $c_s^2 = k_s/m$. Eq. (9) has the form of a Klein–Gordon equation with quadratic and cubic nonlinearities (see Supplementary Note 2.4). Importantly, such equation admits solitary wave solutions of the form[25]

$$u = H \left[ 1 \pm \tanh \left( \frac{x - ct}{w} \right) \right], \tag{10}$$

where $w$ is the width of the propagating pulses.

Next, we determine $c$ and $w$ as a function of the geometry of the system and the energy supplied to the first shell to initiate the pulse. To begin with, we substitute the solution Eq. (10) into Eq. (7) and find that the latter is identically satisfied only if

$$w = \sqrt{\frac{2(c_0^2 - c^2)}{c_s^2}}. \tag{11}$$

Then, we calculate the total energy carried by the transition wave defined by Eq. (10)

$$\begin{aligned} E &= \int_{-\infty}^{\infty} \left[ \frac{1}{2} m \left( \frac{\partial u}{\partial t} \right)^2 + \frac{1}{2} k \left( \frac{\partial^2 u}{\partial x^2} \right) + U(u) \right] dx \\ &= H^2 \left[ \frac{2}{3w} (k + mc^2) + \frac{1}{3} w k_s \right]. \end{aligned} \tag{12}$$

Since in the absence of dissipation $E$ is equal to the energy supplied to the first unit to initiate the pulse, $E_{in}$, we find that

$$H^2 \left[ \frac{2}{3w(c)} (k + mc^2) + \frac{1}{3} w(c) k_s \right] = E_{in}, \tag{13}$$

which we can numerically solve to obtain $c$ for a given $E_{in}$. Further, to obtain an explicit expression for $c$ as a function of $E_{in}$, we take a Taylor's series expansion of Eq. (13) around $c/c_0 = 0$ (since in our system $c/c_0 \sim 0.2$), while retaining terms up to the third order. This yields

$$c = \sqrt{2} c_0 \sqrt{\frac{E_{in}}{E_{min}} - 1}, \tag{14}$$

where

$$E_{min} = \frac{2\sqrt{2}}{3} H^2 \sqrt{k_s k}, \tag{15}$$

represents the minimum amount of input energy required to initiate the transition wave. Eq. (14) confirms that the speed of the propagating transition waves can be tuned by modifying the amount of energy supplied to the system.

To assess the validity of the analytical solution, in Fig. 4c–f we compare the evolution of the transition wave velocity $c$ and width $w$ as predicted by our continuum model (lines) and discrete model (triangular markers). In particular, in Fig. 4c and d we consider three arrays all with $L_t = 28$ mm, but made out of shells with $(H, T_{total}) = (12.5, 3.0)$ mm (red), $(15.0, 4.0)$ mm (purple) and $(17.5, 5.0)$ mm (yellow) and report the evolution of $c$ and $w$ as a function of $E_{in}$. Differently, in Fig. 4d and f we investigate the evolution of $c$ and $w$ as a function of $E_{in}$ for arrays realized using shells with $(H, T_{total}) = (15, 4.0)$ mm when we vary $L_t$. Note that in each plot we report two analytical solutions: one in which $c$ is obtained by solving Eq. (13) (solid lines) and one in which $c$ is given by Eq. (14) (dashed lines). As for the numerical results, these are obtained by conducting simulations with $N = 500$ and $\beta = 0$, using Eq. (10) (with $x = 10$ and $c$ varied to tune $E_{in}$) to

prescribe the pole displacement of the first shell and initiate the pulse and numerically evaluating the integral in Eq. (12) to calculate $E_{in}$ (which is equal to the total energy carried by the pulse). We observe good agreement between the predictions of the discrete model and corresponding results from the continuum model with $c$ obtained by solving Eq. (13) for all considered levels of input energy. Differently, when using Eq. (14) to determine $c$ in the continuum model, the analytical solution matches the experimental results only for low input energies, since the assumption $c/c_0 \to 0$ is violated for large enough values of $E_{in}$. Finally, in full agreement with our experimental observations, both our numerical and analytical results indicate that $c$ increases with $E_{in}$ for all considered double shell arrays, whereas the width $w$ decreases.

While Eq. (14) enables us to calculate $c$ as a function of the input energy and geometric parameters, it does not capture its experimentally observed reduction during propagation caused by dissipation (see Fig. 3d). To overcome this limitation, we assume linear viscous dissipation with damping coefficient $\beta$ and compute the energy dissipated by each shell in the array upon its inversion as (see Supplementary Note 2.4),

$$E_{damped} = \int_{-\infty}^{\infty} \beta \left( \frac{\partial u}{\partial t} \right)^2 dt = \frac{4\beta c H^2}{w}. \tag{16}$$

By introducing Eq. (11), Eq. (16) can be rewritten as

$$E_{damped} = \frac{2\sqrt{2}\beta H^2 c_s c}{\sqrt{c_0^2 - c^2}}, \tag{17}$$

which, by taking a Taylor's series expansion around $c/c_0 = 0$ and retaining terms up to the second order, can be further simplified to

$$E_{damped} \approx \frac{2\sqrt{2}\beta H^2 c_s c}{c_0}. \tag{18}$$

Finally, introduction of Eq. (14) into Eq. (18) yields

$$E_{damped} = 4\beta H^2 c_s \sqrt{\frac{E_i}{E_{min}} - 1}, \tag{19}$$

where $E_i$ denotes the energy carried by the transition wave when propagating through the $i$-th unit.

Note that Eq. (19) can be used to adjust the velocity and account for the effect of damping in our continuum model. Specifically, focusing on the [$i$]-th shell we calculate $E_i$ by subtracting the energy dissipated in the inversion of the previous $i - 1$ shells from the energy supplied to initiate the pulse and subsequently calculate the adjusted velocity using Eq. (14). In Fig. 5a we focus on an array with $N = 10$ double shells identical to those considered in Fig. 3b, with c and report the evolution of $c$ during propagation for different values of input energy, assuming $\beta = 2.5$ kg/s. Notably, we find that the prediction of the continuum model (dashed lines in Fig. 5) nicely agree with the numerical results (continuum lines) up to the fifth shell for moderate and large values of the input energy. Beyond the fifth shell, the free boundary starts to play an important role and this cannot be captured with our continuum model (since we assume the array to be infinitely long). Once again, we observe that by increasing the amount of energy supplied to the first unit, pulses with higher velocity are initiated. However, irrespectively of $E_{in}$, for the level of dissipation present in our structure all transition waves are found to stop after the inversion of the first few units in the absence of favorable end effects.

Next, we use our analytical model to predict the finite propagation distance in systems with a nonzero dissipation.

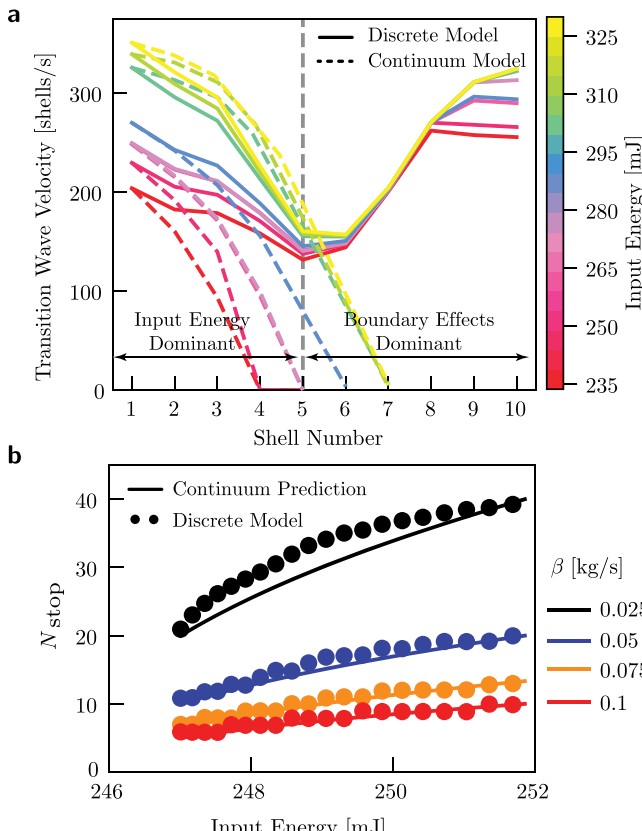

**Fig. 5 Effect of dissipation. a** Comparison of the continuum and discrete model predictions for the transition wave velocity as a function of the propagation distance for an array of 10 double shells with $R = 25.4$ mm, $H = 15$ mm, $T_{total} = 4$ mm and $\beta = 2.5$ kg/s. **b** Theoretical (solid lines, Eq. (24)) and discrete (markers) model predictions for the number of shells flipped before the transition wave stops $N_{stop}$ as a function of the input energy provided to an array of 500 double shells with $R = 25.4$ mm, $H = 15$ mm, $T_{total} = 4$ mm, for different levels of viscous dissipation $\beta$. Note that the levels of dissipation investigated in **b** are much lower than that considered in **a**.

Towards this end, we impose conservation of energy

$$E_{i+1} - E_i = -E_{damped} = -4\beta H^2 c_s \sqrt{\frac{E_i}{E_{min}} - 1}. \qquad (20)$$

To solve Eq. (20) and determine the number of units that the wave switches before stopping, $N_{stop}$, we take the continuum limit of Eq. (20),

$$\frac{dE}{\sqrt{\frac{E}{E_{min}} - 1}} = -4\beta H^2 c_s d\tilde{x}, \qquad (21)$$

where $E(\tilde{x})$ is a continuum function that interpolates $E_i$ as

$$E(\tilde{x} = i) = E_i. \qquad (22)$$

By integrating both sides of Eq. (21) we obtain

$$2E_{min}\sqrt{\frac{E}{E_{min}} - 1}\,\Bigg|_{E_0}^{E_{N_{stop}}} = -4\beta H^2 c_s N_{stop}, \qquad (23)$$

Since $E_0 = E_{in}$ and $E_{N_{stop}} = E_{min}$, $N_{stop}$ can be solved from Eq. (23)

as (see Supplementary Note 2.4)

$$N_{stop} = \frac{E_{min}}{2\beta H^2 c_s}\sqrt{\frac{E}{E_{min}} - 1}. \qquad (24)$$

In Fig. 5b we consider an array comprising 500 double shells with $R = 25.4$ mm, $H = 15$ mm, $T_{total} = 4$ mm and report the evolution of $N_{stop}$ as predicted by Eq. (24) and by our discrete model for different values of $\beta$. We find excellent agreement between analytical and numerical results, with $N_{stop}$ that monotonically increases as either the damping coefficient and the energy input become larger.

**Shells with tunable strain energy landscape.** So far we have shown via a combination of experiments and analyses that a system comprising an array of universally bistable shells separated by air cavities supports the propagation of bidirectional transition waves with characteristics that can be tuned by varying both geometric parameters and the amount of energy supplied to initiate them. However, our results also indicate that the propagation of these pulses in real systems is heavily obstructed by unavoidable dissipation. Motivated by this limitation, we design shells with tunable strain energy landscape and demonstrate that their strategic placement within the array can successfully extend the propagation distance of the waves in dissipative systems. Even though several strategies have been proposed to bias the strain energy landscape of bistable structures[21,26], the approach presented here results in bistable shells with energy landscape that can be easily and actively tuned without the need for further assembly or fabrication. Our tunable shells comprise a double bistable shell (shown in green in Fig. 6a) encapsulated between two single shells (shown in purple in Fig. 6a, see Supplementary Note 1.5 and Supplementary Figs. 13–15). Note that this fabrication process results in the formation of two inflatable cavities (see Fig. 6b). Importantly, the control of their volume enables us to modify on the fly the strain energy landscape of the shell. To demonstrate the concept, in Fig. 6c we consider a tunable shell realized using two shells with $H/R = 0.59$, $T/R = 0.0395$, and $R = 25.4$ mm as caps and a double shell with $H/R = 0.59$, $T_{total}/R = 0.158$ and $R = 25.4$ mm made out of a stiffer silicone rubber (Elite Double 32, Zhermack—see Supplementary Note 1.5). To characterize the static behavior of this shell, we conduct inflation and deflation at different levels of pre-inflation for the two internal cavities. Specifically, in our first test both internal cavities are empty ($\Delta V_{p,1} = \Delta V_{p,2} = 0$), whereas in the second one we pre-inflate one cavity with 10 ml of water ($\Delta V_{p,1} = 0$, $\Delta V_{p,2} = 10$ ml) and in the third one we further add another 10 ml of water to the pre-inflated cavity ($\Delta V_{p,1} = 0$, $\Delta V_{p,2} = 20$ ml). We find that for $\Delta V_{p,1} = \Delta V_{p,2} = 0$ the pressure-volume curve of our tunable shell (see Fig. 6c) is qualitatively identical to the one of the double shell (see Fig. 2g). Differently, when one of the internal cavities is pre-inflated (i.e., $\Delta V_{p,2} \neq 0$) the maximum pressure required to invert the tunable shell during inflation drops, whereas the magnitude of the negative pressure required to bring it back to its original state increases. This indicates that the pre-inflation of an internal cavity increases the elastic strain energy stored in the initial state, but simultaneously decreases that associated to the inverted configuration. As a result, only a small input pressure is required to invert a tunable shell with a pre-inflated internal cavity and such inversion leads to the release of a large amount of energy. Finally, we note that, by pre-inflating the other internal cavity (i.e., $\Delta V_{p,1} \neq 0$, $\Delta V_{p,2} = 0$) we can decrease the elastic energy stored in the initial state and increase that associated to the inverted configuration, thus realizing a shell that release a large amount of energy when snapping back to the initial state.

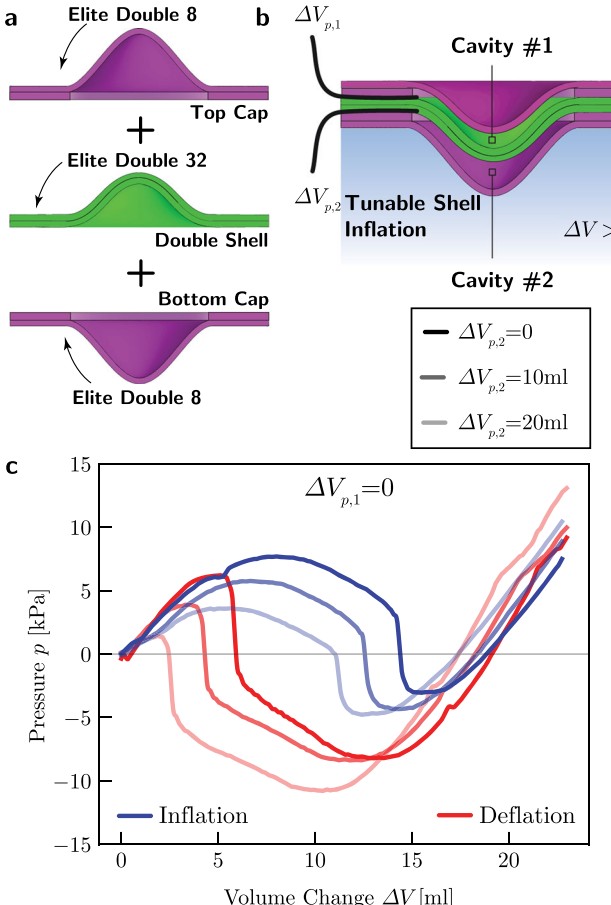

**Fig. 6 Shells with tunable energy landscape. a** Our tunable shell comprises a double shell and two single shell caps. **b** Geometry of the tunable shell. **c** Experimental pressure-volume relationship for the tunable shell during its quasi-static inflation (blue) and deflation (red) for 3 different levels of pre-inflation for an internal cavity.

To test the capability of such tunable shells to extend the propagation distance of transition waves in dissipative systems, we consider an array with $N = 12$ double shells identical to the ones used in the experiments of Fig. 3. Since dissipation prevents the transition waves to switch all elements in the array when those are simple double shells (see blue markers in Fig. 3d), we place our tunable shell in between the 6th and 7th shell of the array, as shown in Fig. 7a–b. First, we charge the tunable shell by pre-inflating the cavity that faces the end to which the pressure pulse is applied with 20 ml of water (see Fig. 7a,b). Then, we initiate a pulse by applying an input pressure $\Delta p = 69$ kPa to either the first or last unit in the array. Remarkably, we find that the charged tunable shell enables the transition wave to propagate through the entire array of 12 double shells (see Fig. 7c,d), since the energy that it releases when snapping to its inverted state compensates for the energy lost by the pulse because of dissipation. Furthermore, we emphasize that the introduction of the tunable shell in the array does not inhibit the bidirectionality of the supported transition waves (see Fig. 7c,d). This is because by simply changing the polarity of the tunable shell (i.e., pre-inflating the opposite cavity), we can reverse the direction in which energy will be released. Finally, we note that the control of the tunable shell by adding or removing volume to one of its internal cavities is extremely simple, and does not require re-assembly of the array.

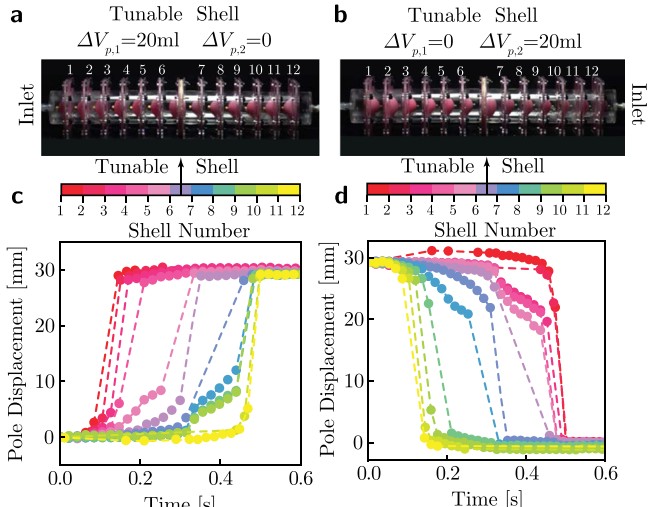

**Fig. 7 Transition waves in arrays of shells with tunable energy landscape. a**, **b** Arrays of 12 double shells, where a charged tunable shell is placed between the 6th and 7th shell of the array. **c**, **d** Experimental pole displacement histories for each shell in the array upon propagation of a transition waves initiated by supplying $\Delta p = 69$ kPa of pressure for 100 ms. Vertical black arrows point to the shell number and location of the charged tunable shell in the array.

## Discussion

In summary, we have demonstrated a robust strategy for the design of doubly curved thick shells which are bistable for any combination of geometric parameters. Further, we have studied the propagation of transition waves in 1D arrays of such shells coupled by compressible fluid cavities and demonstrated that the supported pulses are bidirectional. Our combined experimental, numerical and analytical results reveal that the characteristics of the supported non-linear waves can be tuned not only by altering the geometry of the system but also by controlling the amount of energy supplied to initiate them. However, since our universally bistable shells do not release energy when transitioning between their two stable states, the distance traveled by the supported transition waves is limited by unavoidable dissipative phenomena. To compensate for this without sacrificing bidirectionality, we designed thick bistable shells with tunable energy landscape. We then demonstrated that their strategic placement in 1D shell arrays can extend the propagation distance of transition waves, since they can be easily set to release the energy required to compensate for dissipation. As such, by combining universally bistable and tunable shells we realized 1D arrays that support the bidirectional propagation of transition waves over finite distances while being easy to reset and tune.

Even though in this study we used rigid chambers to connect adjacent shells, we envision the proposed strategy to provide a new route for soft robotic locomotion. By making the chambers unidirectionally stretchable, they would sequentially extend during the propagation of transition waves and emulate the rectilinear locomotion of snakes. In addition, our system's unique property, namely the dependence of transition wave velocity to the input energy, could enable the design of smart energy absorption devices which effectively transfer energy but are able to avoid energy concentrations through dissipation. Further, systems based on our strategy could also serve as energy sensors, as the energy input can be determined by monitoring the effective transition wave velocity.

Finally, we believe that the proposed strategies to design bistable doubly curved shells have the potential to impact

applications that extend beyond transition waves, including soft mechanical logic gates and reconfigurable structures.

## Methods
Details on the geometry, design, fabrication, testing, and Finite Element modeling of the doubly curved shells, universally bistable shells and shells with tunable energy landscape are provided in Supplementary Note 1. The full details for the experimental setup, as well as for the testing and modeling of transition waves in 1D arrays of bistable doubly curved shells are provided in Supplementary Note 2.

## Data availability
The experimental and numerical data in support of the findings in this study are available from the corresponding author upon request.

## Code availability
All numerical codes used to computationally study the propagation of transition waves and all Abaqus Python scripts used to create the FE models are available from the corresponding author upon request.

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

## Acknowledgements
K.B. acknowledges support from NSF under grants DMR-1420570 and DMR-1922321 and Army Research Office under grant W911NF-17-1-0147. The authors acknowledge Dr. James Weaver for 3D printing the molds used to cast all shells.

## Author contributions
N.V., B.D, B.G., and K.B. designed research; N.V. and B.D. analyzed data; N.V., B.D., and B.G. performed experiments; B.G. designed 3D printed molds used to cast the shells. B.D. developed the continuum model; N.V. and K.B. wrote the paper.

## Competing interests
The authors declare no competing interests.
