## [Peer Review File · Nature Communications]

This article presents a combined theoretical and experimental study of the propagation of snap-through waves within a new kind of meta-material. The authors develop an experimental system that consists of several bistable elastic caps that influence one another by displacing the gas between them. This is possible because they have found a clever way to fabricate caps that remain bistable, even as the thickness changes. There are several good ideas in this paper and the analysis presented is convincing (though I have a couple of suggestions on this below). The one real weakness in the paper overall is that no compelling application of these ideas is presented. If more could be said about how this sort of system could be used in, for example, soft robotics, then it would be much more compelling for a journal like *Nature Communications*.

My main scientific comment is that the authors have not fully exploited their continuum theory. In particular, equation [13] can be simplified for small wave speed, $\frac{c}{c_0} \ll 1$, and this solution gives (I believe) some new insight into the problem. To see this, note that [11] can be written: $w(c) = \sqrt{2} \frac{c_0}{c_s} \left(1 - \frac{c^2}{c_0^2}\right)^{1/2}$ so that [13] can then be written:

$$\frac{3E_{in}}{kH^2} = 2 \left(1 + \frac{c^2}{c_0^2}\right) w(c)^{-1} + \frac{c_s^2}{c_0^2} w(c) = \frac{\sqrt{2}c_s}{c_0} \left[\left(1 + \frac{c^2}{c_0^2}\right) \left(1 - \frac{c^2}{c_0^2}\right)^{-\frac{1}{2}} + \left(1 - \frac{c^2}{c_0^2}\right)^{1/2} \right]$$

Expanding the RHS for $\frac{c}{c_0} \ll 1$ I find that

$$\frac{3E_{in}}{kH^2} \approx \frac{\sqrt{2}c_s}{c_0} \left(2 + \frac{c^2}{c_0^2}\right)$$

which suggests that:

$$c \approx c_0 \left(\frac{3E_{in}}{\sqrt{2}(k \times k_s)^{1/2}H^2} - 2 \right)^{1/2}$$

There are two features of this result that I think are interesting: Firstly, this result shows that there is a minimum input energy for wave propagation, $E_{min} = \frac{2\sqrt{2}}{3} (k k_s)^{1/2} H^2$. This result seems plausible given the results of the discrete analysis presented in figures 4 c and d: there is indeed a minimum energy for wave propagation and this minimum seems to be strongly dependent on H (fig 4c) and more weakly dependent on the 'air stiffness' k (see fig. 4d). Secondly, this result shows that the speed of propagation increases very rapidly for input energies just above the minimum but then starts to level off a little (also as seen in figs 4c and d). I think it is an important test of the continuum theory to compare the results above (together with the corresponding result for w) against the discrete results (as well as their numerical solution of the continuum problem).

I also think that the above result would give a more compelling picture of how many snaps can be generated before the wave extinguishes itself. At present, eqn (S28) assumes each snapping shell dissipates the same amount of energy. In fact, eqn (S27) shows that this is a sensitive function of c and, given the above result, is sensitively dependent on the rate at

which the energy decreases close to extinction. For example, close to extinction (i.e. using $\frac{c}{c_0} \ll 1$) one could write the energy prior to the $(n+1)^{\text{st}}$ snapping event in terms of that prior to the n^{th} by writing:

$$E_{n+1} = E_n - 4\beta H^2 \frac{c}{w(c)} \approx E_n - 4\beta c_s H^2 \left(\frac{E_n}{E_{min}} - 1 \right)^{1/2}$$

This is a difference equation for E_n and can easily be solved using the discrete-continuum approximation already used by the authors. I find that:

$$4\beta c_s H^2 (n_* - n) = 2E_{min} \left(\frac{E_n}{E_{min}} - 1 \right)^{1/2}$$

for some constant of integration n_* (which corresponds to the number of shells after which extinction occurs). A simple estimate, then, is that extinction should occur after snapping

$$n_* \approx \frac{E_{min}}{2\beta c_s H^2} \left(\frac{E_{in}}{E_{min}} - 1 \right)^{1/2}$$

shells. As a result, inputting more energy has limited efficiency at snapping more elements – since snap-through happens faster, the dissipation is also higher – as reflected by the scaling $n_* \sim E_{in}^{1/2}$.

I think that the results above are important for explaining features of the discrete model but, at least as far as I could tell, are not presented in either the main text or the SI.

I also have a number of minor comments that I think the authors should consider.

Minor comments:

- I found the way that the single shell and double-glued shell were presented was confusing – given that the text builds up the expectation that the shell presented here will be “bistable for any thickness” figure 1c is a bit of a surprise. On further reading, it is clear that this is just a preliminary figure, and should be contrasted with the behavior of the double-glued shell presented in figure 2. It would be helpful if there could be clearer contrast between figures 1b and 2e and 1c and 2f in the text, and more anticipation that the single shell structure is *not* the one used throughout the paper. One way to do this might be to combine the relevant panels together in a single figure and present them side-by-side. Alternatively, better sign-posting in the text would help.
- The discussion at the bottom of the second page regarding the inflation of an individual double shell is unclear. Was the shell first attached to a rigid cylinder into which the water was pumped? Was the water pumped back between the constituent shells?

- I am confused about the speed of the transition wave presented in fig. 3d (and elsewhere in the SI). From fig. 3b and c, it seems that the wave snaps 10 shells in a period of 0.2s. That would make the average wave speed around 50 shells/s, which is significantly smaller than the 200-300 shells/s reported in the y-axis of fig. 3d. Is there an issue here?
- The authors mention repeatedly that the wave speeds up again as it approaches the free end. However, I could not see a clear discussion of how this softening enters into the discrete model (though it is clearly there since fig. 3d shows the same phenomenon in the discrete model results).
- The speed c_s is only defined after eqn [11], even though it appears in eqn [9]. It would make sense to define it earlier.
- The results in figure 4 (and the discussion after eqn [13]) are, I think, for the case of zero dissipation. This should be specified clearly in the caption of figure 4.
- The authors state that their experiments indicate that the transition wave velocity increases with E_{in} but I could not see this experimental data in either the main text or SI. The only indication is the limited data (two different pressures) in fig. 3. It would be good to see more experimental evidence of this – the model is clear, but since this is a main finding of the theory, it is important to compare this back to as much experimental data as possible.

Reviewer #2 (Remarks to the Author):

This paper describes a process by which symmetrically-bistable thick shells are fabricated from asymmetrically-bistable shells by first displacing them to a flat deformed configuration and then adhering them. The resulting symmetrically-bistable structures are then assembled in 1-D arrays at it is shown that by snapping through one bistable shell the resulting pressure change in the adjacent cavity causes the adjacent cell to itself snap through and so on until dissipation effects start to dominate. In a final step a concept is presented in which the shells can be made asymmetrically bistable. This asymmetry allows energy showed in the less preferential stable state to be released as useful work during its transition to the more preferential stable state with the result that the propagation can be made to extend further.

I should say first of all that I very much enjoyed reading this paper. However, I have the following comments for the authors' consideration.

The generation of symmetrically bistable structures via the coupling of asymmetrically-bistable structures is clever, and I believe this is a novel and useful contribution. I have concerns about the description of such structures as universal, however. I have considered the behaviour of coupled von Mises trusses with biasing springs as a useful simplified analogue (see attached figure). In this case it can be seen that as the degree of asymmetric bistability of the sub-structures increases, and beyond when the second stable state has been annihilated, the locations of the stable states of the coupled system move further away from the neutrally-stable point and the stiffness greatly increases. If we consider that this behaviour will be replicated by the coupled shells we can see that the theoretically-stable locations move to a degree of displacement which cannot be comfortably attained by the structure (at least without significant higher-order deformations). I do not know what would happen to the stability landscape in this case — it would be interesting to investigate — but I suspect that in the practical limit the adhered configuration (Fig. 2c) will become the preferred stable configuration. It is also the case that there is practical limit on how far the initial thick shell can be compressed without the formation of local buckling etc. At best the universality can surely only be claimed for the theoretical and not the practical response of the structure.

The analysis of the response of the array is well carried out and the results are plausible and well validated. It is interesting to see the effect of the propagation of the instability. From a fundamental perspective the behaviour seems quite straightforward (essentially it is summed up by Fig. 4b) although I agree there is value to investigating a practical implementation.

I am not persuaded that the final section adds much to the story. It has been shown in literature (e.g. [https://doi.org/ 10.1115/1.4000417](https://doi.org/10.1115/1.4000417)) that asymmetric bistability can be utilized for unilateral

high-frequency actuation. The practical implementation that is presented is interesting but it seems that it is one of several possible techniques that can be used to add a bias to the energy landscape.

Minor points:

In the abstract it is stated that increased bending stiffness causes a stable energy state to annihilate as the thickness becomes larger. I think this is over-simplified — although the increased thickness leads to increased bending stiffness, increasing bending stiffness alone does not necessarily cause stable states to disappear.

Also in the abstract the phrase “bistable for any thickness” falls under my first point above.

Double curved and doubly curved appear to be used interchangeably.

In the conclusion the phrase “in vivo” would usually be restricted to operations carried out on living entities

Response to the Referees

Universally Bistable Shells with Nonzero Gaussian Curvature for Two-Way Transition Waves

Manuscript number: NCOMMS-20-25334

Nikolaos Vasios^a, Bolei Deng^a, Benjamin Gorissen^a, and Katia Bertoldi^{a,b,c,*}

^aJ.A. Paulson School of Engineering and Applied Sciences, Harvard University, Cambridge, MA 02138, USA; ^cKavli Institute for Bionano Science and Technology, Harvard University, Cambridge, MA 02138, USA

In this document, we provide a copy of the comments and points raised by each reviewer and address them one at a time. Pages R1-R14 address the comments raised by Reviewer 1, whereas pages R15-R21 address the comments raised by Reviewer 2. A copy of the reviewer's text is provided for each comment. Changes to the main text or Supporting information are highlighted in blue. All modified figures are included as part of our response for completeness.

Response to Referee #1

This article presents a combined theoretical and experimental study of the propagation of snap-through waves within a new kind of meta-material. The authors develop an experimental system that consists of several bistable elastic caps that influence one another by displacing the gas between them. This is possible because they have found a clever way to fabricate caps that remain bistable, even as the thickness changes. There are several good ideas in this paper and the analysis presented is convincing (though I have a couple of suggestions on this below). The one real weakness in the paper overall is that no compelling application of these ideas is presented. If more could be said about how this sort of system could be used in, for example, soft robotics, then it would be much more compelling for a journal like Nature Communications.

We thank the reviewer for the positive remarks and insightful suggestions. The point raised by the reviewer is a valid one and we modified the main text to illustrate potential applications of our strategy. More specifically, we have added the following text on page 8 of the manuscript

“Even though in this study we used rigid chambers to connect adjacent shells, we envision the proposed strategy to provide a new route for soft robotic locomotion. By making the chambers unidirectionally stretchable, they would sequentially extend during the propagation of transition waves and emulate the rectilinear locomotion of snakes. Additionally, our system's unique property, namely the dependence of transition wave velocity to the input energy, could enable the design of smart energy absorption devices which effectively transfer energy but are able to avoid energy concentrations through dissipation. Further, systems based on our strategy could also serve as energy sensors, as the energy input can be determined by monitoring the effective transition wave velocity.”

My main scientific comment is that the authors have not fully exploited their continuum theory. In particular, equation [13] can be simplified for small wave speed, $c/c_0 \ll 1$ and this solution gives (I believe) some new insight into the problem. To see this, note that [11] can be written $w(c) = \sqrt{2} \frac{c_0}{c_s} \left(1 - \frac{c^2}{c_0^2}\right)^{1/2}$ so that [13] can be written,

$$\frac{3E_{in}}{kH^2} = 2 \left(1 + \frac{c^2}{c_0^2}\right) w(c)^{-1} + \frac{c_s^2}{c_0^2} w(c) = \frac{\sqrt{2}c_s}{c_0} \left[\left(1 + \frac{c^2}{c_0^2}\right) \left(1 - \frac{c^2}{c_0^2}\right)^{-1/2} + \left(1 - \frac{c^2}{c_0^2}\right)^{1/2} \right]$$

Expanding the RHS for $c/c_0 \ll 1$, I find that,

$$\frac{3E_{in}}{kH^2} \approx \sqrt{2} \frac{c_s}{c_0} \left(2 + \frac{c^2}{c_0^2}\right),$$

which suggests that,

$$c \approx c_0 \left(\frac{3E_{in}}{\sqrt{2}k \times k_s H^2} - 2 \right)^{1/2}$$

There are two features of this result that I think are interesting: Firstly, this result shows that there is a minimum input energy for wave propagation,

$$E_{min} = \frac{2\sqrt{2}}{3} \sqrt{k \times k_s} H^2$$

this result seems plausible given the results of the discrete analysis presented in figures 4c and d: there is indeed a minimum energy for wave propagation and this minimum seems to be strongly dependent on H (fig 4c) and more weakly dependent on the ‘air stiffness’ k (see fig. 4d). Secondly, this result shows that the speed of propagation increases very rapidly for input energies just above the minimum but then starts to level off a little (also as seen in figs 4c and d). I think it is an important test of the continuum theory to compare the results above (together with the corresponding result for w) against the discrete results (as well as their numerical solution of the continuum problem). I also think that the above result would give a more compelling picture of how many snaps can be generated before the wave extinguishes itself. At present, eqn (S28) assumes each snapping shell dissipates the same amount of energy. In fact, eqn (S27) shows that this is a sensitive function of c and, given the above result, is sensitively dependent on the rate at which the energy decreases close to extinction. For example, close to extinction (i.e. using $c/c_0 \ll 1$ one could write the energy prior to the $(n+1)^{st}$ snapping event in terms of that prior to the n^{th} by writing:

$$E_{n+1} = E_n - 4\beta H^2 \frac{c}{w(c)} \approx E_n - 4\beta c_s H^2 \left(\frac{E_n}{E_{min}} - 1 \right)^{1/2}$$

This is a difference equation for E_n and can easily be solved using the discrete-continuum approximation already used by the authors. I find that:

$$4\beta c_s H^2 (n^* - n) = 2E_{min} \left(\frac{E_n}{E_{min}} - 1 \right)^{1/2}$$

for some constant of integration n^* (which corresponds to the number of shells after which extinction occurs). A simple estimate, then, is that extinction should occur after snapping

$$n^* = \frac{E_{min}}{2\beta c_s H^2} \left(\frac{E_n}{E_{min}} - 1 \right)^{1/2}$$

shells. As a result, inputting more energy has limited efficiency at snapping more elements – since snap-through happens faster, the dissipation is also higher – as reflected by the scaling $n^* \sim E_{in}^{1/2}$. I think that the results above are important for explaining features of the discrete model but, at least as far as I could tell, are not presented in either the main text or the SI.

We thank the reviewer for the suggestions and detailed derivations that he/she was willing to provide. We consider this discussion to be an extremely valuable addition to our work. To this end, we have updated both the main text and SI to include these derivations and the new analytical results. More specifically, we have modified Fig. 4 (also shown below as Fig. R1 for completeness) and added the following text to the manuscript to emphasize that an explicit expression for c can be obtained

"Since in the absence of dissipation E is equal to the energy supplied to the first unit to initiate the pulse, E_{in} , we find that

$$H^2 \left[\frac{2}{3w(c)}(k + mc^2) + \frac{1}{3}w(c)k_s \right] = E_{\text{in}}, \quad [\text{R1}]$$

which we can numerically solve to obtain c for a given E_{in} . Further, to obtain an explicit expression for c as a function of E_{in} , we take a Taylor's series expansion of Eq. (R1) around $c/c_0 = 0$ (since in our system $c/c_0 \sim 0.2$), while retaining terms up to the third order. This yields

$$c = \sqrt{2}c_0 \sqrt{\frac{E_{\text{in}}}{E_{\text{min}}} - 1}, \quad [\text{R2}]$$

where

$$E_{\text{min}} = \frac{2\sqrt{2}}{3}H^2 \sqrt{k_s k}, \quad [\text{R3}]$$

represents the minimum amount of input energy required to initiate the transition wave. Eq. (R2) confirms that the speed of the propagating transition waves can be tuned by modifying the amount of energy supplied to the system. To assess the validity of the analytical solution, in Figs. R1c-f we compare the evolution of c and w as predicted by our continuum model (lines) and discrete model (triangular markers). In particular, in Figs. R1c and d we consider three arrays all with $L_t = 28$ mm, but made out of shells with $(H, T_{\text{total}}) = (12.5, 3.0)$ mm (red), $(15.0, 4.0)$ mm (purple) and $(17.5, 5.0)$ mm (yellow) and report the evolution of c and w as a function of E_{in} . Differently, in Figs. R1d and f we investigate the evolution of c and w as a function of E_{in} for arrays realized using shells with $(H, T_{\text{total}}) = (15, 4.0)$ mm when we vary L_t . Note that in each plot we report two analytical solutions: one in which c is obtained by solving Eq. (R1) (solid lines) and one in which c is given by Eq. (R2) (dashed lines). As for the numerical results, these are obtained by conducting simulations with $N = 500$ and $\beta = 0$, using Eq. (10) (with $x = 10$ and c varied to tune E_{in}) to prescribe the pole displacement of the first shell and initiate the pulse and numerically evaluating the integral in Eq. (12) to calculate E_{in} (which is equal to the total energy carried by the pulse). We observe good agreement between the predictions of the discrete model and corresponding results from the continuum model with c obtained by solving Eq. (R1) for all considered levels of input energy. Differently, when using Eq. (R2) to determine c in the continuum model, the analytical solution matches the experimental results only for low input energies, since the assumption $c/c_0 \rightarrow 0$ is violated for large enough values of E_{in} . Finally, in full agreement with our experimental observations, both our numerical and analytical results indicate that c increases with E_{in} for all considered double shell arrays, whereas the width w decreases. "

Further, we have added a new panel to Fig. 5 (also shown below as Fig. R2 for completeness) and added the following text to describe how the number of units that the wave switches before stopping can be determined

"By introducing Eq. (11), Eq. (6) can be rewritten as

$$E_{\text{damped}} = \frac{2\sqrt{2}\beta H^2 c_s c}{\sqrt{c_0^2 - c^2}}, \quad [\text{R4}]$$

which, by taking a Taylor's series expansion around $c/c_0 = 0$ and retaining terms up to the second order, can be further simplified to

$$E_{\text{damped}} \approx \frac{2\sqrt{2}\beta H^2 c_s c}{c_0}. \quad [\text{R5}]$$

Finally, introduction of Eq. (R2) into Eq. (R5) yields

$$E_{\text{damped}} = 4\beta H^2 c_s \sqrt{\frac{E_i}{E_{\text{min}}} - 1}, \quad [\text{R6}]$$

where E_i denotes the energy carried by the transition wave when propagating through the i -th unit.

.....

Next, we use our analytical model to predict the finite propagation distance in systems with a nonzero dissipation. Towards this end, we impose conservation of energy

$$E_{i+1} - E_i = -E_{\text{damped}} = -4\beta H^2 c_s \sqrt{\frac{E_i}{E_{\text{min}}} - 1}. \quad [\text{R7}]$$

To solve Eq. (R7) and determine the number of units that the wave switches before stopping, N_{stop} , we take the continuum limit of Eq. (R7),

$$\frac{dE}{\sqrt{\frac{E}{E_{\text{min}}} - 1}} = -4\beta H^2 c_s d\bar{x}, \quad [\text{R8}]$$

Fig. R1. Analytical and numerical results in the absence of dissipation. (a) Schematic of our system. (b) Discrete model used to represent the response of our system. (c)-(d) Influence of the input energy provided to initiate the pulse on (c) the pulse velocity, c , and (d) the pulse width, w , for three shell geometries with $(H, T_{\text{total}}) = (12.5, 3)$ mm (yellow), $(H, T_{\text{total}}) = (15, 4)$ mm (blue) and $(H, T_{\text{total}}) = (17.5, 5)$ mm (red) and $R = 25.4$ mm, as predicted by the discrete (markers) and continuum models (lines). (e) Wave velocity, c , and (f) width, w , vs. input energy, E_{in} , for an array of universally bistable shells with $(H, T_{\text{total}}) = (15, 4)$ mm and $R = 25.4$ mm, for three values of shell-to-shell spacing, $L_t = 32$ mm (yellow), 28 mm (blue) and 22 mm (red), as predicted by the discrete (markers) and continuum model (lines). Note that in (c)-(f) we report two analytical solutions: one in which c is obtained by solving Eq. (R1) (solid lines) and one in which c is given by Eq. (R2) (dashed lines).

where $E(\tilde{x})$ is a continuum function that interpolates E_i as

$$E(\tilde{x} = i) = E_i. \quad [\text{R9}]$$

By integrating both sides of Eq. (R8) we obtain

$$2E_{\text{min}} \sqrt{\frac{E}{E_{\text{min}}} - 1} \Big|_{E_0}^{E_{N_{\text{stop}}}} = -4\beta H^2 c_s N_{\text{stop}}, \quad [\text{R10}]$$

Since $E_0 = E_{\text{in}}$ and $E_{N_{\text{stop}}} = E_{\text{min}}$, N_{stop} can be solved from Eq. (R10) as (see Supporting Information)

$$N_{\text{stop}} = \frac{E_{\text{min}}}{2\beta H^2 c_s} \sqrt{\frac{E}{E_{\text{min}}} - 1}. \quad [\text{R11}]$$

In Fig. R2b we consider an array comprising 500 double shells with $R = 25.4$ mm, $H = 15$ mm, $T_{\text{total}} = 4$ mm and report the evolution of N_{stop} as predicted by Eq. (R11) and by our discrete model for different values of β . We find excellent agreement between analytical and numerical results, with N_{stop} that monotonically increases as either the damping coefficient and the energy input become larger. "

Fig. R2. Effect of dissipation. (a) Comparison of the continuum and discrete model predictions for the transition wave velocity as a function of the propagation distance for an array of 10 double shells with $R = 25.4$ mm, $H = 15$ mm, $T_{\text{total}} = 4$ mm and $\beta = 2.5$ kg/s. (b) Theoretical (solid lines, Eq. R11) and discrete (markers) model predictions for the number of shells flipped before the transition wave stops N_{stop} as a function of the input energy provided to an array of 500 double shells with $R = 25.4$ mm, $H = 15$ mm, $T_{\text{total}} = 4$ mm, for different levels of viscous dissipation β .

Detailed derivations of these new analytical results have been included in SI.

Fig. R3. Our shells. (a) Shell geometry. (b) Elastic strain energy landscape as a function of the pole displacement during the quasi-static inflation/deflation of two shells with $H/R = 0.59$ and $T/R = 0.0787$ (red) and $T/R = 0.1653$ (blue). (c) Evolution of the energy release U_r upon inversion as a function of H/R and T/R . (d) Flattening of two identical single shells. (e) Gluing the two single shells in the flat deformed configuration to obtain the double shell. (f) The geometry of the double shell. (g) Strain energy landscape for double shells with different thickness. (h) Contour plot of the energy release U_r as a function of H and T_{total} .

Comment 1B

I found the way that the single shell and double-glued shell were presented was confusing – given that the text builds up the expectation that the shell presented here will be “bistable for any thickness” figure 1c is a bit of a surprise. On further reading, it is clear that this is just a preliminary figure, and should be contrasted with the behavior of the double-glued shell presented in figure 2. It would be helpful if there could be clearer contrast between figures 1b and 2e and 1c and 2f in the text, and more anticipation that the single shell structure is not the one used throughout the paper. One way to do this might be to combine the relevant panels together in a single figure and present them side-by-side. Alternatively, better sign-posting in the text would help

This is another good point. Following the reviewer’s suggestion we have combined relevant panels of Figs. 1 and 2 to present the single and double-glued shells side by side. Panels a, b and c of the new Fig. 1 (also shown as Fig. R3 for completeness) showcase the geometry and response upon inflation of the single doubly-curved thick shell. Panels d, e and f illustrate the step by step process to construct the universally bistable doubly-curved double shells. Finally panels g and h showcase the response of the doubly-curve universally bistable shells upon inflation, in direct contrast to panels b and c for the single shell.

Further, we modified Fig.2 of the main text to serve as the experimental validation of our numerical approach for modeling the behavior of the doubly-curved universally bistable shells. Panel a of the new Fig. 2 (also shown below as Fig. R4 for completeness) showcases a schematic of the testing apparatus used to inflate and deflate the shells in quasi-static conditions using a syringe pump. Panels b and c show a comparison between the experimental findings and numerical predictions for the pressure-volume and the pole displacement-volume relationships of the universally bistable shells.

Finally, we have added a new Figure (Figure S6 - also shown below as Fig. R5 for completeness) to clarify how the shells were inflated/deflated.

Fig. R4. Experimental characterization of our universally bistable thick shells. (a) Schematic of the experimental setup used to quasi-statically inflate and deflate the universally bistable shells using water, while being submerged in a water tank. (b)-(c) Quasi-static pressure-volume and pole displacement-volume relationships obtained upon inflation and deflation of a double shell with $H/R = 0.59$ and $T_{total}/R = 0.158$ (with $R=25.4$ mm) in experiments (dashed lines) and FE simulations (solid lines).

Fig. R5. Experimental setup for the quasi-static testing of shells subjected to inflation and deflation. (a) The custom apparatus comprised of clear cast acrylic tube and laser-cut acrylic plates. (b) The syringe pump used to inflate/deflate she shells while fully submerged in the water tank. (c) Schematic of the experimental setup used to quasi-statically inflate and deflate the universally bistable shells using water, while being submerged in a water tank.

Comment 1C

The discussion at the bottom of the second page regarding the inflation of an individual double shell is unclear. Was the shell first attached to a rigid cylinder into which the water was pumped? Was the water pumped back between the constituent shells?

In an effort to better explain our experimental process and avoid confusion we added panel a to Fig. 2 of the main text (also shown as Fig. R4 for completeness). In the panel we illustrate the experimental apparatus used to quasi-statically inflate our universally bistable shells. Further, we have also re-worded the text in page 2 of our main text to clearly illustrate our experimental process:

"We then characterize its quasi-static response by attaching its boundaries to an enclosed rigid cylinder and supplying water with a syringe pump (Pump 33DS, Harvard Apparatus) at a constant rate of 30 mL/min to inflate it and deflate it (see Fig. R4a)."

Comment 1D

I am confused about the speed of the transition wave presented in fig. 3d (and elsewhere in the SI). From fig. 3b and c, it seems that the wave snaps 10 shells in a period of 0.2s. That would make the average wave speed around 50 shells/s, which is significantly smaller than the 200-300 shells/s reported in the y-axis of fig. 3d. Is there an issue here?

We thank the reviewer for pointing out this inconsistency between the wave speeds reported in Fig. 3d and those that one can compute from the data reported in Fig. 3b and Fig. 3c. The reason for this inconsistency was that velocities reported in Fig. 3d were computed by determining the value of c that best fits the experimental data for the pole displacement of the shells using the analytical solution

$$u(t) = H \left[1 \pm \tanh \left(\frac{x - ct}{\sqrt{\frac{2(c_0^2 - c^2)}{c_s^2}}} \right) \right]. \quad [\text{R12}]$$

However, for our highly dissipative system, the value of c obtained using Eq. (R12) (which inherently neglects any dissipation effects) seems to overestimate the actual transition wave velocity. As such, we have recomputed all transition wave velocities reported in Fig. 3d by estimating the time t_i at which shell i is half-inverted at $u(t_i) = H$ (i.e. this is done by determining the zero of the tanh argument in Eq. (R12)). Then, the transition wave velocity (in shells/s) between shells i and $i + 1$ is computed as

$$c_i \approx \frac{1}{t_{i+1} - t_i}. \quad [\text{R13}]$$

In Fig. 3d we now report the transition wave velocities for all experiments and discrete model simulations obtained using Eq. R13. We find that the recomputed wave velocities are much closer to ~ 50 shells/s. For completeness we include the modified Fig. 3 below. We also modified the main text to indicate how such velocities are being calculated

"To better characterize these elastic waves, in Fig. 3d we report the evolution of their velocity (calculated by monitoring the time at which $u_{pole,i} = H$) during propagation."

Finally, we note that in the absence of dissipation (or for systems with small dissipation) the two methods described above for determining the transition wave velocity (i.e. from a linear squares fit using Eq. (R12) or by using Eq. (R13)) produce almost identical predictions.

Fig. R6. Bidirectional transition waves in 1D arrays of bistable shells connected with compressible fluid cavities. (a) Schematic of the 1D array. (b)-(c) Bidirectional propagation of transition waves in an array of 10 universally bistable shells with $H = 15$ mm, $R = 25.5$ mm and $T_{total} = 4$ mm, excited by supplying $\Delta p = 69$ kPa of pressure for 100 ms. (d) Evolution of the transition wave velocity during propagation for an array of 10 universally bistable shells excited at the left (red markers) and right (pink markers) ends by applying a pressure $\Delta p = 69$ kPa for 100 ms. Blue markers represent the velocity for an identical pulse propagating in an array of $N = 12$ universally bistable shells, whereas green and yellow markers correspond to the wave velocity for a pulse excited using $\Delta p = 172$ kPa and a pulse in an array with reduced shell to shell spacing ($L_t = 22$ mm), respectively.

Comment 1E

The authors mention repeatedly that the wave speeds up again as it approaches the free end. However, I could not see a clear discussion of how this softening enters into the discrete model (though it is clearly there since fig. 3d shows the same phenomenon in the discrete model results).

We thank the reviewer for the comment. The softening enters the discrete model naturally by means of the boundary conditions. During the integration of the discrete model equations, the force component transmitted to each shell due to the compression of the air cavities accounts for the two closest adjacent shells and chambers to the shell for which the computation is carried over. The last shell in the array, only has 1 neighbour and is only coupled to the rest of the chain using only 1 fluid cavity and therefore experiences a reduced force due to air cavity compression. As a result, for some array lengths, when the head of the pulse reaches the end of the array, the energy required to switch the last few units decreases, thereby leading to an increase of the transition wave velocity. To better explain this phenomenon, we have added the following sentence in page 4 of the main text

"On the other hand, when the head of the pulse reaches the end of the array, the energy required to switch the last few units decreases, thereby leading to an increase of the transition wave velocity."

Comment 1F

The speed c_s is only defined after eqn [11], even though it appears in eqn [9]. It would make sense to define it earlier.

We thank the reviewer for pointing out this inconsistency. We have addressed this comment by defining the the speed c_s immediately after its introduction in Eq. (9). Specifically, we have modified the text after Eq. (9) to read as

"By introducing Eq. (8), Eq. (7) simplifies to

$$\frac{\partial^2 u}{\partial \zeta^2} = \frac{c_s^2}{c_0^2 - c^2} u \left(\frac{u}{H} - 1 \right) \left(\frac{u}{H} - 2 \right), \quad [\text{R14}]$$

where $c_s^2 = k_s/m$."

Comment 1G

The results in figure 4 (and the discussion after eqn [13]) are, I think, for the case of zero dissipation. This should be specified clearly in the caption of figure 4.

We thank the reviewer for this comment. We have modified the caption of Fig. 4 to make it clear that all data reported in Fig. 4 are for systems with zero dissipation. The caption of Fig. 4 now reads,

"Fig. 4. Analytical and numerical results in the absence of dissipation. ..."

Comment 1H

The authors state that their experiments indicate that the transition wave velocity increases with E_{in} but I could not see this experimental data in either the main text or SI. The only indication is the limited data (two different pressures) in fig. 3. It would be good to see more experimental evidence of this – the model is clear, but since this is a main finding of the theory, it is important to compare this back to as much experimental data as possible.

We thank the reviewer for this comment. In Fig. S22 of the revised SI (also shown below as Fig. R7 for completeness) we report experimental results and numerical predictions for 8 experiments in which pulses were initiated using 10, 15, 20 and 25 psi of pressure.

Fig. R7. Results of the discrete model and comparison against experiments for forwards and backwards propagation of transition waves initiated by pulses of increasing energy . Solid lines represent the discrete model predictions whereas markers correspond to experimental measurements. (a)-(d) Forward propagation for pulses initiated using 10, 15, 20 and 25 psi of pressure respectively. (e)-(h) Backward propagation for pulses initiated using 10, 15, 20 and 25 psi of pressure respectively.

This paper describes a process by which symmetrically-bistable thick shells are fabricated from asymmetrically- bistable shells by first displacing them to a flat deformed configuration and then adhering them. The resulting symmetrically-bistable structures are then assembled in 1-D arrays at it is shown that by snapping through one bistable shell the resulting pressure change in the adjacent cavity causes the adjacent cell to itself snap through and so on until dissipation effects start to dominate. In a final step a concept is presented in which the shells can be made asymmetrically bistable. This asymmetry allows energy showed in the less preferential stable state to be released as useful work during its transition to the more preferential stable state with the result that the propagation can be made to extend further. I should say first of all that I very much enjoyed reading this paper. However, I have the following comments for the authors' consideration.

We thank the reviewer for his/her comments. We are happy to see that the reviewer enjoyed reading our paper.

Comment 2A

The generation of symmetrically bistable structures via the coupling of asymmetrically-bistable structures is clever, and I believe this is a novel and useful contribution. I have concerns about the description of such structures as universal, however. I have considered the behaviour of coupled von Mises trusses with biasing springs as a useful simplified analogue (see attached figure). In this case it can be seen that as the degree of asymmetric bistability of the substructures increases, and beyond when the second stable state has been annihilated, the locations of the stable states of the coupled system move further away from the neutrally-stable point and the stiffness greatly increases. If we consider that this behaviour will be replicated by the coupled shells we can see that the theoretically-stable locations move to a degree of displacement which cannot be comfortably attained by the structure (at least without significant higher-order deformations). I do not know what would happen to the stability landscape in this case — it would be interesting to investigate — but I suspect that in the practical limit the adhered configuration (Fig. 2c) will become the preferred stable configuration. It is also the case that there is practical limit on how far the initial thick shell can be compressed without the formation of local buckling etc. At best the universality can surely only be claimed for the theoretical and not the practical response of the structure.

We thank the reviewer for this comment. We believe that the our shells are bistable as long as the resulting geometry can still be identified as a thick shell. Increasing the thickness arbitrarily, at some point, results in a structure that can no longer be referred to as a shell but rather resembles a 3D solid. In such cases, our findings do not apply. From a theoretical standpoint, any thick shell constructed using our approach is bistable. However, it is true that for practical applications extremely thick shells are challenging to fabricate and test.

Comment 2B

The analysis of the response of the array is well carried out and the results are plausible and well validated. It is interesting to see the effect of the propagation of the instability. From a fundamental perspective the behaviour seems quite straightforward (essentially it is summed up by Fig. 4b) although I agree there is value to investigating a practical implementation.

To illustrate potential applications of our strategy, we have added the following text on page 8 of the manuscript

"Even though in this study we used rigid chambers to connect adjacent shells, we envision the proposed strategy to provide anew route for soft robotic locomotion. By making the chambers unidirectionally stretchable, they would sequentially extend during the propagation of transition waves and emulate the rectilinear locomotion of snakes. Additionally, our system's unique property, namely the dependence of transition wave velocity to the input energy, could enable the design of smart energy absorption devices which effectively transfer energy but are able to avoid energy concentrations through dissipation. Further, systems based on our strategy could also serve as energy sensors, as the energy input can be determined by monitoring the effective transition wave velocity."

Comment 2C

I am not persuaded that the final section adds much to the story. It has been shown in literature (e.g. <https://doi.org/10.1115/1.4000417>) that asymmetric bistability can be utilized for unilateral high-frequency actuation. The practical implementation that is presented is interesting but it seems that it is one of several possible techniques that can be used to add a bias to the energy landscape.

We thank the reviewer for the comment. To acknowledge the fact that several strategies have been proposed to bias the strain energy landscape of bistable structures we have added the following sentence to the main text

"Even though several strategies have been proposed to bias the strain energy landscape of bistable structures, the approach presented here results in bistable shells with energy landscape that can be easily and actively tuned without the need for further assembly or fabrication."

Furthermore, we believe, that our strategy is the first that enables the continuous modification of the strain energy profile while being able to maintain the systems bi-directionality. Specifically, by inflating the left cavity of the tunable shell we extend the propagation distance when the wave is travelling "left-to-right". Similarly, simply by deflating the left cavity and inflating the right cavity of the tunable shell by the same amount, we can now extend the propagation distance when the wave is travelling "right-to-left". Notably, the deflation and inflation of the tunable shell's cavities can be achieved without modifying or disassembling our array.

Comment 2D

In the abstract it is stated that increased bending stiffness causes a stable energy state to annihilate as the thickness becomes larger. I think this is over-simplified — although the increased thickness leads to increased bending stiffness, increasing bending stiffness alone does not necessarily cause stable states to disappear.

We thank the reviewer for this comment. We agree that this statement is an oversimplification of the nonlinear mechanics involved in the inversion of our doubly-curved shells. In fact, bending stiffness is a quantity relevant about the base (i.e. undeformed) configuration of the shells and does not necessarily affect the shells' bistability. We have therefore modified the first sentence of the abstract to read as

"Multi-welled energy landscapes arising in shells with nonzero Gaussian curvature typically fade away because of the increased bending energy required for inversion."

Comment 2F

Also in the abstract the phrase “bistable for any thickness” falls under my first point above.

We thank the reviewer for this comment. We believe that our response to comment 2A provides a valid response to this comment as well.

Comment 2G

Double curved and doubly curved appear to be used interchangeably.

We thank the reviewer for this comment. We have addressed all instances in which “double curved” appeared in the text and replaced them with “doubly curved”.

Comment 2H

In the conclusion the phrase “in vivo” would usually be restricted to operations carried out on living entities

We thank the reviewer for the comment and for pointing out this error. We have replaced the word “in vivo” with the word “actively”.

REVIEWERS' COMMENTS

Reviewer #1 (Remarks to the Author):

The authors have carefully considered my comments and made appropriate changes to the manuscript as a result. In particular, I think the discussion of the minimum energy for propagation and the behavior with wave speed are a significant improvement. I also think that the discussion of possible applications at the end is sufficient.

I have a few minor comments that I believe should be addressed before the paper is published, but they are minor:

1. In their plotting of N_{stop} the authors seem to show that the analytical formula based on the argument I provided (and they confirmed) works very well in comparison to the numerics. This seems surprising at first (given the difference seen in figure 5a). This is simply because the simulations reported in Fig 5 b were performed on arrays with a very large number of shells, while those in a were not. In my view, this should be spelled out more explicitly in the paragraph following eqn [24].

(As a minor comment on this plot, I was also confused that the value of beta used in (b) seems to be 100 times smaller than that in (a). Is there a reason for this, or is it a typo?)

2. The new figure 1 is a big improvement on the former, but I think may still be hard to understand at first glance. Would it be possible to note in the caption that (a)-(c) refer to single shells while (d)-(h) refer to double shells? If not, perhaps a label to the side of (a) and (d)-(f) could emphasize the important difference?

Dominic Vella

Response to the Referees

Universally Bistable Shells with Nonzero Gaussian Curvature for Two-Way Transition Waves

Manuscript number: NCOMMS-20-25334

Nikolaos Vasios^a, Bolei Deng^a, Benjamin Gorissen^a, and Katia Bertoldi^{a,b,c,*}

^aJ.A. Paulson School of Engineering and Applied Sciences, Harvard University, Cambridge, MA 02138, USA; ^cKavli Institute for Bionano Science and Technology, Harvard University, Cambridge, MA 02138, USA

In this document, we provide a copy of the comments and points raised by the reviewer and address them one at a time. A copy of the reviewer's text is provided for each comment. Changes to the main text or Supporting information are highlighted in blue. All modified figures are included as part of our response for completeness.

Response to Referee #1

The authors have carefully considered my comments and made appropriate changes to the manuscript as a result. In particular, I think the discussion of the minimum energy for propagation and the behavior with wave speed are a significant improvement. I also think that the discussion of possible applications at the end is sufficient.

I have a few minor comments that I believe should be addressed before the paper is published, but they are minor:

We are glad to see that the reviewer finds the revised version of the manuscript significantly improved. In the following we address his/her new minor remarks.

Comment 1A

In their plotting of N_{stop} the authors seem to show that the analytical formula based on the argument I provided (and they confirmed) works very well in comparison to the numerics. This seems surprising at first (given the difference seen in figure 5a). This is simply because the simulations reported in Fig 5b were performed on arrays with a very large number of shells, while those in a were not. In my view, this should be spelled out more explicitly in the paragraph following eqn [24].

(As a minor comment on this plot, I was also confused that the value of beta used in (b) seems to be 100 times smaller than that in (a). Is there a reason for this, or is it a typo?)

As correctly pointed out by the reviewer, the agreement between the analytical prediction for N_{stop} and the numerical results is much better in Fig. 5b than in Fig 5a. However, this is not caused by the number of shells in the array, but rather by the level of damping. When comparing numerical results with the analytical solution, in order to get a good agreement the influence of damping should be such that it does not influence the shape of the solution. Unfortunately, for the level of damping present in our experiments (i.e. $\beta = 2.5$ kg/s) this is not the case and therefore the agreement is not very good. Differently, in Fig. 5b we consider lower values of damping and we find very good agreement between the analytical solution and the numerical results. To clarify this point we have added the following text to the caption of Fig. 5:

"Note that the levels of dissipation investigated in (b) are much lower than that considered in (a). "

Comment 1B

The new figure 1 is a big improvement on the former, but I think may still be hard to understand at first glance. Would it be possible to note in the caption that (a)-(c) refer to single shells while (d)-(h) refer to double shells? If not, perhaps a label to the side of (a) and (d)-(f) could emphasize the important difference?

We are glad to see that the reviewer find the revised version of Fig. 1 more clear and informative. Following his/her suggestion, we have modified the caption to explicitly state that (a)-(c) refer to single shells while (d)-(h) refer to double shells.